# The Stability and Properties of Polystyrene/Kaolinite Nanocomposites during Synthesis via Emulsion Polymerization

**DOI:** 10.3390/polym15092094

**Published:** 2023-04-28

**Authors:** José Costa de Macêdo Neto, Bruno Mello de Freitas, Adalberto Gomes de Miranda, Reinaldo de Almeida Rodrigues, Gilberto Garcia Del Pino, Antônio Claudio Kieling, Marcos Dantas dos Santos, Sergio Duvoisin Junior, Antônio Estanislau Sanches, Israel Gondres Torné, Cláudia Cândida Silva, João Carlos Martins da Costa, Roger Hoel Bello

**Affiliations:** Department of Materials Engineering, School of Engineering, Amazonas State University, Avenida Darcy Vargas, 1200, Parque Dez de Novembro, Manaus 69850-000, AM, Brazil

**Keywords:** emulsion, polymerization, nanocomposites, kaolinite

## Abstract

The aim of this work was to study the stability and morphological properties of polystyrene latex containing kaolinite as a filler during the process of synthesis of nanocomposites viaemulsion polymerization. Nanocomposites with 1, 3, and 5 wt% of kaolinite were prepared. Latexes with 1 to 3 wt% of kaolinite were stable during the polymerization reaction. Hydrodynamic diameters of 93.68 and 82.11 nm were found for latexes with 1 and 3 wt% of kaolinite, respectively. The quantities of 1 to 3 wt% of kaolinite added during the reaction did not influence the reaction conversion curves or the number of particles. X-ray diffraction (XRD) and unconventional techniques of scanning electron microscopy (SEM) and high-resolution transmission electron microscopy (HRTEM) showed the presence of exfoliated and intercalated structures of the kaolinite.

## 1. Introduction

Polymer nanocomposites are a class of materials that consists of nanoparticles dispersed in apolymer matrix.In order to be considered nanoparticles, at least one dimension has to be between 10 and 100 nm [1]. Materials used as nanoparticles in polymer nanocomposites can benanospheres (for example, TiO_2_, with diameters between 50 and 200 nm and Au, whose diameter is around 36–148 nm [2,3]), nanotubes (for example, halloysite, whosediameter is in the range of 30–100 nm) and nanofibers (for example, carbon fibers, exhibitan average diameter of 3.2 nm),and natural and synthetic clays [4,5,6].

Examples of synthetic clays used in polymer nanocomposites are layered double hydroxides (LDH) (basal spacing between 0.77 and 2.57 nm) [7,8].The clay mineral group is made up of montmorillonite, illite (basal spacing: 9.87 nm), vermiculite (basal spacing: 1.45–1.55 nm), and kaolinite (basal spacing: 0.72 nm) [6,9,10,11].Polymeric nanocomposites using clays as nanoreinforcement usually have better mechanical properties when compared to polymers without nanoparticles [12]. The improvement of these properties is due to the high aspect ratio of the nanoclaysand their morphology and distribution in the polymeric matrix [13].

When using mineral clays as nano-reinforcement to produce polymeric nanocomposites, the compatibility between the polymer and the clay is fundamental in order to achieve the desired propertiesand stable latex. In this sense, the clay usually needs to undergo some type of pre-treatment before being added, which consists of the intercalation of a substance, such as dimethylsulfoxide (DMSO), 6-aminohexane acid (AHA), N-methylformamide, carboxylic acid, and functionalized ammoniaand urea, in the clay [14,15,16,17]. The amount and the typeof organic modifier can have a direct effect on the stability of the elastic material [18].

Aside fromcompatibility, a good dispersion of the clays in the matrix is desirable because, on a nanometric scale, it leads to a better interaction between the clay and the polymer. The morphology of the clays inside the polymer matrix can be either tactoid (agglomerates), intercalated (aggregated),orexfoliated (dispersed). The amount of intercalated and exfoliated clay interacting with the polymer chains is responsible for the improvement in the mechanical, thermal, and barrier properties, but an exfoliated morphology is preferred because it gives the best mechanical properties [19]. Other factors that can affect the stability of the latex are the amount of clay used and the presence of polymer-stabilizing agents [20].

Concerning the methods for polymeric nanocomposite synthesis, the most used are melt blending and in situ polymerization [21,22]. In situ emulsion polymerization is nature-friendly because it synthesizes water-based polymers and forms films that can be applied in automotive, textile, leather, wood, paint, and other fields [23]. Emulsion polymerization using clays as a filler offers advantages over conventional emulsion polymerization, such as improved mechanical [24], thermal [25], and optical [26] properties of the resulting films. In the literature, mineral clays, such as montmorillonite, laponite, graphite, and LDH, have been used as a filler to produce nanocompositesusing in situ emulsion polymerization [9,24,25,26]. However, only a few studies deal with kaolinite, and they tend to consider other polymers, such as polyacrylonitrile and polymethylmethacrylate [26,27].

In this paper, we studied the production of polystyrene nanocomposites filled with kaolinite using in situ emulsion polymerization and investigated the influence of kaolinite in the colloidal stability on the styrene polymerization reaction. The stability of the latex during the polymerization reaction is important in order to obtain a product with defined physical and mechanical properties, such as glass transition temperature and tensile strength. To the best of our knowledge, this is the first work to report the behavior of a polymer/kaolinite hybrid latex during polymerization.

## 2. Experimental

### 2.1. Synthesis of Nanocomposites

The desired amount of 1, 3, and 5 wt% of modified kaolinite was added to 133.90 g of styrene and dispersed using a magnetic stirrer at 20 rpm for 2 h at room temperature. Subsequently, the solution was sonicated for 24 min. After stirring the suspended system of St monomer and Kao-DMSO, 537.94 g of water, an aqueous solution with 3.48 g of SLS, the clay, and an aqueous solution of 0.57 g of KPS were poured (in this order) into a1-L batch reactor equipped with a heating jacket, mechanical stirrer, reflux condenser, and purging tube.

The stirring speed and nitrogen flux in the reactor were 60 rpm and 5 L/min, respectively. The reaction time was 90 min. To obtain samples in powder form for further characterization, the volatile liquids were evaporated in beakers in a laminar flow cabinet and dried in an oven at 100 °C for 4 h. The conversion was obtained via gravimetry. A total of 1, 3, and 5 wt% kaolinite–DMSO nanocomposites (weight percent of clay based on monomer weight) were synthesized and named: 1%-Kao-DMSO-PS, 3%-Kao-DMSO-PS, and 5%-Kao-DMSO-PS, respectively. Polystyrene samples are referred to as PS throughout the text. The residues that were generated in the latex were dried for 48 h at 70 °C and weighed on a semi-analytical scale (Mars, AS1000C, Säo Paulo, Brazil).

### 2.2. Characterization

The X-ray diffraction analysis (XRD) ofthe nanocomposites was performed in a diffractometer (Shimadzu XRD 7000) with CuKα radiation, k = 1.54060 Å with 2θ varying between 1.4 and 70°. The XRD for the kaolinite was performed in a diffractometer (Philips Analytical X-ray) with CuKα, k = 1.54060 Å with 2θ varying between 5.0 and 70.0°.Hydrodynamic size and zeta potential measurements were performed in a multi-purpose titrator (Malvern, MPT-2) using dynamic light scattering (DLS).

The Fourier transform infrared (FTIR) spectra were recorded in the range of 400–4500 cm^−1^ at room temperature using aspectrometer (PerkinElmer, Spectrum One-FTIR) and KBr pellets. This technique was used to indicate the presence of kaolinite in the nanocomposites.

Scanning electron microscopy (SEM) images obtained in an electron microscope (LEO 440i) were used to observe the interaction between the kaolinite and the latex. The morphology of the nanocomposites was also observed using a high-resolution transmission electron microscope (HRTEM) (JEOL JEM 3010 URP) equipped with a CCD camera (Gatan MSC794) and the Gatan Digital Micrograph software 1.5.0.0. The nanocomposite samples were macerated to obtain a thin and homogeneous powder, which was added and mixed into a solution of epoxy resin and a catalyst. An ultramicrotome (Leica, UltraCutUCT, Säo Paulo, Brazil) was used to obtain thin sections (about 120 nm in thickness) of the nanocomposite samples. The sections were then placed on a carbon film-coated copper grid of 3 mm indiameter.

## 3. Results and Discussion

### 3.1. Molecular Intercalation

Figure 1a,b exhibits the XRD pattern of pure kaolinite and the DMSO intercalated kaolinite. According to Figure 1, the basal spacing for the pure kaolinite related to the plane (001), peak 2θ = 12.38°, was 0.72 nm (d (001)). For the kaolinite intercalated with DMSO (Figure 1b), the basal spacing was found to be higher, in this case, 1.12 nm (peak 2θ = 7.9°).

Figure 2a,b show the structure of the kaolinite before and after the intercalation with DMSO. Figure 2a shows that the kaolinite is composed of silicate layers bonded by weak ionic bonds between the oxygen from the layer formed by the SiO_4_ tetrahedron and the hydroxyl group from the Al_2_(OH)_6_ octahedron layer and maintains a neutral structure [28].

Figure 2b shows the expansion of the basal spacing of the kaolinite when intercalated with DMSO molecules. This expansion is related to the molecular configuration in the form of a bilayer of the DMSO molecule [28,29]. The intercalation occurred because the DMSO molecule broke the internal hydrogen bonds between the lamellae of the kaolinite, which connected the hydroxyls of the octahedral layer of the clay, as shown in Figure 2b [29,30,31].

As the DMSO is a polar molecule, breaking the connection between the lamellae of kaolinite becomes easier. In the intercalated structure, there was a reaction between the sulfoxide group (=S=O) of DMSO, which has a positive character, and the internal hydroxyl of the octahedral layer of kaolinite, which has a negative character, which left the methyl groups to the side of the tetrahedral lamellae after the reaction [14,29]. In addition, after the treatment with DMSO, the mineral clays became organophilic [14].

Figure 3a,b show the FTIR spectra of the kaolinite and the DMSO intercalated kaolinite, respectively. In Figure 3a, the bands observed at 3696 cm^−1^ and 3622 cm^−1^ can beattributed to the stretching vibration of the hydroxyl groups [28,30]. Figure 3b shows the reduction of the band at 3696 cm^−1^ and the formation of new bands at 3670, 3552, and 3527 cm^−1^, which can be attributed to hydrogen bonding between the DMSO and the kaolinite [28,30].

### 3.2. Particle Diameter

Figure 4 shows the particle size distribution inthe polymerization of styrene. Samples were collected at 5, 10, 20, 40, 60, and 90 min. As can be seen at 5 and 10 min, a large size distribution and an average size of 52.9 and 60.4 nm, respectively, were observed. After 20 min of reaction, narrower distributions and greater sizes occurred. The largest particle size (90.39 nm) was found at 90 min. Figure 4 also shows that at 5 min, there is a bimodal curve. The first one can be attributed to the polymer particles, and the second one can be related to the monomer droplets that disappear during the course of the reaction. The extinction of the curve of the monomer droplets at 10 min indicates that the monomers migrated from the droplets to the interior of the micelles while the reaction occurred [32]. Additionally, in Figure 4, the absence of particles with dimensions of 2–10 nm (size range that is characteristic of the micelles) suggests a fast migration from the monomer to the micelles, thus causing fast growth of the polymer particles [33,34].

Figure 5 shows the particle size distribution for the polymerization of 1%Kao-DMSO-PS latex. The largest average size of the polymer particles (93.68 nm) occurred after 90 min of reaction. The increase in kaolinitecontent led to an increase in the size of the nanocomposite particles in the latex [35]. The results showed that the addition of 1 wt% of organically modified kaolinite increased the particle size of the polystyrene.

Figure 6 shows the latex particle sizedistribution for 3%Kao-DMSO-PS latex. This sample exhibited an average particle diameter (82.11 nm) that was smaller than the average size of the latex particles in 1%Kao-DMSO-PS (93.68 nm) (Figure 5) and in pure polystyrene (90.39 nm) (Figure 4) after90 min of reaction. During the polymerization reaction, the addition of 3% clay influenced the solubility of the emulsifier and affected the critical micelle concentration (CMC), and this influence on the CMC reduced the average particle size. This reduction in particle size was attributed to the interaction between the surface of the clay and the polymer particle due to the increase in clay content to 3 wt%. The literature also states that the location of the particles on the surface of kaolinite lamellae may have reduced the interaction between the PS particles and water and thereby reduced the particle size. The adsorption of the kaolinite particles at the emulsion droplet interface further reduces the interfacial tension between the oil and water phases. Facilitating the breakdown of large droplets will result in smaller droplets [33,34,35,36].

### 3.3. Conversions

Figure 7 illustrates the conversion curves over time for styrene (PS) without reinforcement (PS) and forthe nanocomposite with 1, 3, and 5% of kaolinite (1, 3, and 5%Kao-DMSO-PS).In the reaction time of up to 50 min, it was observed (Figure 7) that the PS has a higher conversion than the nanocomposite. In this initial interval, the radical diffusion formed in the aqueous phase may have had its flow toward the inside of the micelles made more difficult by the presence of the lamellae of the kaolinite. The lamellae of kaolinite may have acted as a physical barrier to themonomers that migrate towards the micelles and thus retard the conversion forthe nanocomposites 1%Kao-DMSO-PS, 3%Kao-DMSO-PS, and 5%Kao-DMSO-PS in phase I of the reaction.Such behavior has been reported in other studies [14,35].

It is observed that, within 50 min of reaction, the 3%Kao-DMSO-PS presented a higher conversion than 1%Kao-DMSO-PS. In their studies, Moghadam and Moghbeli (2016) [36] and Nistor et al. (2015) [37] reportthat the clay lamellae may have become a locus of polymerization for monomers with the existence of several micelles. As the 3%Kao-DMSO-PS has a greater amount of clay, it is possible that a higher nucleation rate may have occurred if compared to 1%Kao-DMSO-PS.

At60 min, 1%Kao-DMSO-PS showed a higher conversion than PS and 3%Kao-DMSO-PS. This behavior becomes constant until the end of the reaction. An increase in the amount of clay to 3%Kao-DMSO-PS reduced the solubility of the emulsifier in water and, consequently, reduced the critical micellar concentration (CMC). The increased amount of clay provides a maximum CMC, which generates a maximum polymer particle. From a certain amount of clay, which can be between 1 and 3% in the system, the emulsifier may not cover the entire surface of the particles anymore, thus leading to a possible increase of clots and reduced conversion, andworks in the literature also report such phenomena [34,35,37].

For a nanocomposite with 5% kaolinite (5%Kao-DMSO-PS), it can beobserved that the conversion was lower during the reaction in relation to the 1%Kao-DMSO-PS, 3%Kao-DMSO-PS, and thePS. At the end of the reaction, a large clot formationwas observed, which precipitated atthe bottom of the reactor. According toNistor et al. (2015) [37], the increase in the amount of clay may have reduced the CMC and increased the number of micelles. Thus, there may have been an increase in polymer particle nucleation, and this may have affected the reaction.

Between 50 and 60 min, the nanocomposites 1%Kao-DMSO-PS and 3%Kao-DMSO-PS showed a peak conversion, with 3%Kao-DMSO-PS with a higher peak. The clay was adsorbed on the latex particles and, as the clay has a high aspect ratio (length/thickness), it acted as an alternative reaction locus for the growth of the polymer particles, which caused a conversion peak [33]. After 70 min, 1% Kao-DMSO-PS showed a higher conversion than PS and 3%Kao-DMSO-PS. This behavior becomes constant until the end of the reaction. Although the 3%Kao-DMSO-PS nanocomposite presented a higher peak, the amount of kaolinite in the 3%Kao-DMSO-PSnanocomposite hindered the diffusion of molecules to the particle, which resulted in a lower final conversion.

### 3.4. Study of the Residues Obtained in the Synthesis

After the polymerization reactions with kaolinite, the presence of residues was observed. This subsection provides a study of these residues, as well as their formation. For this analysis, we used X-ray diffraction (XRD) and scanning electron microscopy (SEM). The influence of these residues on the kinetics of the reactions was also studied.

Table 1 shows quantities of the residues obtained from the polymerization reactions. In Table 1, it can be seen that the kaolinite intercalated with only DMSO (Kao-DMSO-PS) shows residues only for the nanocomposite with 5% kaolinite (5%Kao-DMSO-PS).

The treatment of the kaolinite with DMSO made it organophilic, and this may explain the absence of residues in the production of the 1%Kao-DMSO-PS and 3%Kao-DMSO-PSnanocomposites. In this case, the kaolinite treated with DMSO may have been dispersed in the monomer. The formation of residues in the nanocomposite with 5%Kao-DMSO-PS may be due to the formation of clots. A discussion with more details is provided in Section 3.5,which includes the SEM and XRD analysis of the residues obtained.

Figure 8 shows the XRD nanocomposite residue obtained from the reaction with 5%Kao-DMSO-PS. In the figure, it is possible to observe the absence of the characteristic peaks of kaolinite; however, Figure 8 shows the presence at 19.25° of an amorphous halo that is characteristic of polystyrene. Thus, it suggests that the residue obtained from the polymerization of the 5%Kao-DMSO-PS nanocomposite is from clots. According to Yang et al. (2012) [28], the quantity of clots increases proportionally with the amount of clay in the emulsion polymerization. Tang et al. (2018) [29] also observed an increase in clots with the addition of clay tothe emulsifier system. According to Nistor et al. (2015) [37], the increase in the amount of clay may have reduced the CMC (critical micelle concentration) and increased the number of micelles. Thus, there may have been an increase in polymer particle nucleation, which may have affected the reaction.

Figure 9 shows the image of the residue obtained from the reaction of 5%Kao-DMSO-PS. It is observed that the residue has a homogeneous shape, which suggests that it is due to the clots that were formed.

### 3.5. XRD and HRTEM of the Nanocomposites

In Figure 10b,c, it can be observed that the XRD patterns of the nanocomposites with 1 and 3% of kaolinite are like the patterns for pure PS diffraction. According to Yang et al. (2012) [28], this similarity suggests that this clay exfoliated in the polymeric matrix because it does not show a peak indicating the presence of the clay structure. The absence of peaks related to the plane (001) indicates that the lamellar structure was destroyed when the growth of the polymer chains between the lamellae of kaolinite occurred. According to Tang et al. (2018) [29], the increase in the size of the polymer chains between the lamellae of kaolinite can increase the interlayer spacing of the clay, and thus XRD peaks relating to the interlayer spacing are no longer detected. For Neto et al. (2015) [35], the breaking up of the lamellar structure of the kaolinite may have been facilitated by the interleaving of the styrene monomer between the lamellae of the kaolinite after the swelling step and the ultrasound, which forced the monomer interlayer between the lamellae of the clay.

Although XRDis an effective method for determining the intercalation and exfoliation in nanocomposites, nothing can be said about the distribution of the clay in the polymer matrix. Thus, it is difficult to reach a conclusion regarding the mechanism of the nanocomposite formation and its morphology based only on XRD. According to Yang et al. (2009) [38], analysis using HRTEM allows a better understanding of the morphology and dispersion of clay via the image obtained. According to Nistor et al. (2015) [37], TEM and XRD techniques are complementary for characterizing the morphology of nanocomposites.

Figure 11 shows the 3%Kao-DMSO-PS nanocomposite using HRTEM with a magnification of 200,000×, which is observed in intercalated clay and exfoliated form in the polystyrene matrix. According to Tu et al. (2008) [39], the monomer would have spread between the clay lamellae and polymerized, thus resulting in interleaving. However, for Ammala et al. (2007) [40], the high cohesive energy between the lamellae also may have contributed to an interleaved morphology. According to Prabawa et al. (2020) [41], the morphology of exfoliated kaolinite appears as dark lines distributed randomly in the polymer, and this means that the clay completely lost itsorder among the lamellae [42]. In Figure 11, it is observed that the exfoliated lamellae are shown as having an arc-shaped morphology. According to Zatta et al. (2011) [4], the deformation of the lamellae induced by the monomer polymerization interleaved between the two lamellae is transmitted throughout the entire lamellae, causing contraction of one of the lamellae, thus resulting in an arc-shaped or anchor morphology in one of the clay lamellae.

According to Yang et al. (2009) [38], the good dispersion of the clay can also be attributed to the initial state polymerization, wherein the viscosity of the reactive medium is relatively low, which facilitates the dispersion of the clay. The agitation during the polymerization and the acquisition of stable latex may also have contributed to the good dispersion of clay.

## 4. Conclusions

The XRD showed that by using DSMO, it was possible to perform molecular intercalation.The addition of 3% modified clay influenced the particle size during polymerization. The addition of 1% modified kaolinitedid not influence particle size during polymerization. The use of 1% kaolinite was found to be the ideal amount since it has greater conversion, and the particle size and diameter distribution are close to those of clay-free polystyrene latex.The addition of 5% modified kaolinite destabilized the polymerization system and reduced the conversion, which resulted in the appearance of residues. Via XRD analysis of the nanocomposites with 1 and 3% modified kaolinite, it was possible to produce polymer nanocomposites, andvia HRTEM analysis, it was possible to observe intercalated and exfoliated kaolinite.

## Figures and Tables

**Figure 1 polymers-15-02094-f001:**
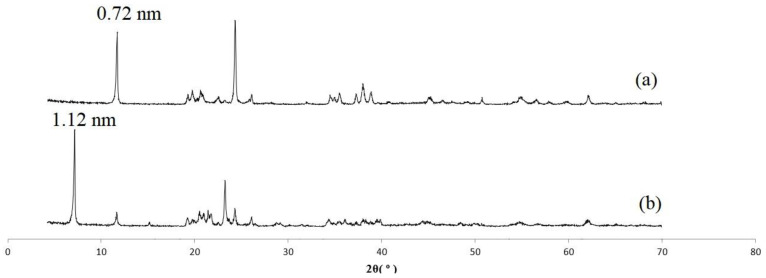
XRD patterns. (**a**) Kaolinite and (**b**) Kaolinite–DMSO.

**Figure 2 polymers-15-02094-f002:**
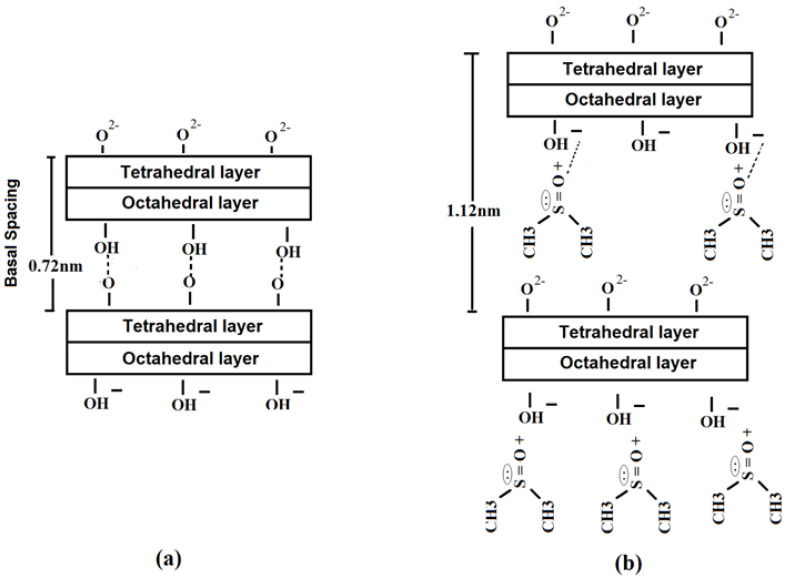
Scheme for the preparation of the kaolinite: (**a**) Kaolinite, (**b**) Kaolinite–DMSO.

**Figure 3 polymers-15-02094-f003:**
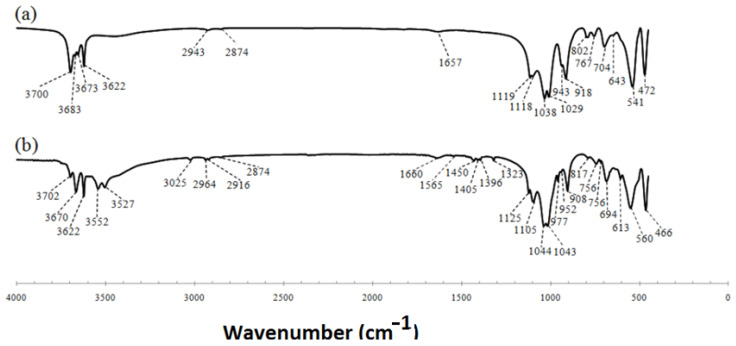
(**a**) Kaolinite and (**b**) Kaolinite–DMSO.

**Figure 4 polymers-15-02094-f004:**
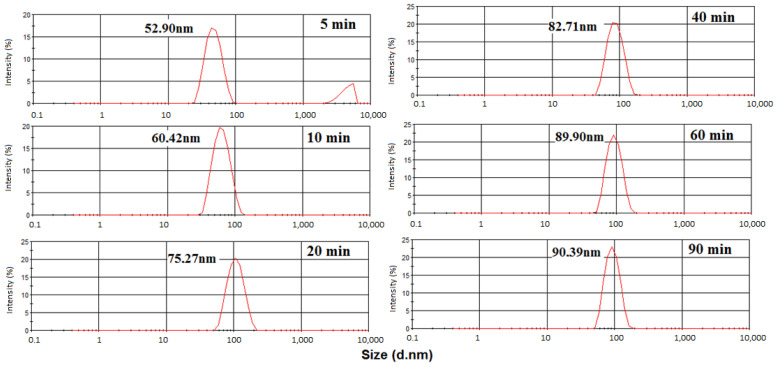
Particle size distributions in the polymerization of polystyrene latex.

**Figure 5 polymers-15-02094-f005:**
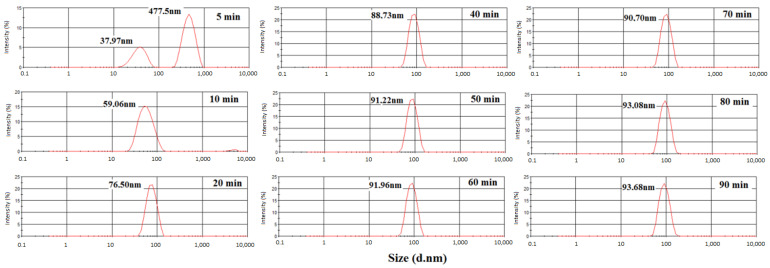
Particle size distributions of 1%Kao-DMSO-PS latex.

**Figure 6 polymers-15-02094-f006:**
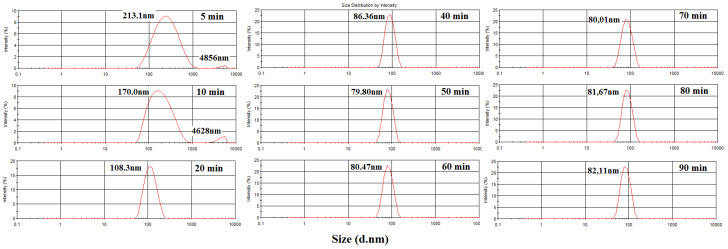
Particle size distributions of 3%Kao-DMSO-PS latex.

**Figure 7 polymers-15-02094-f007:**
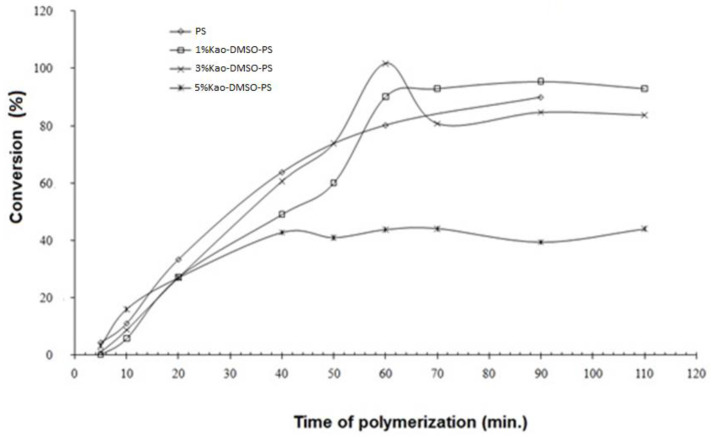
Conversion vs. time profile for the polymerization of pure styrene (PS) and styrene with the addition of modified kaolinite (Kao-DMSO-PS) in the proportions of 1, 3, and 5% by weight.

**Figure 8 polymers-15-02094-f008:**
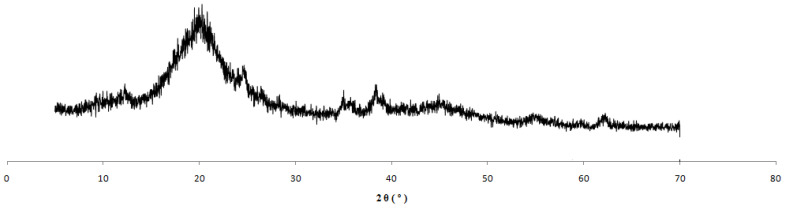
XRD of the residues from the reaction of 5%Kao-DMSO-PS.

**Figure 9 polymers-15-02094-f009:**
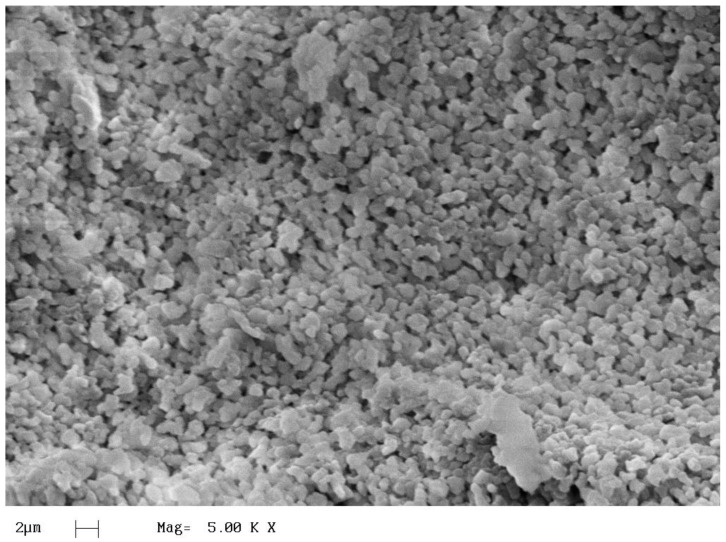
SEM of the residues found in 5%Kao-DMSO-PS.

**Figure 10 polymers-15-02094-f010:**
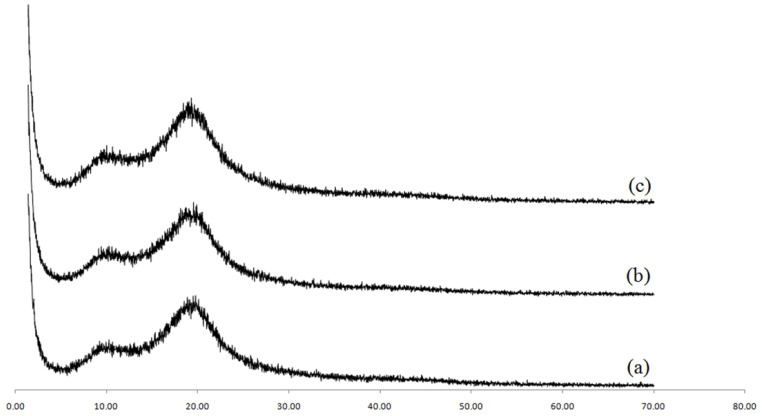
XRD patterns of (**a**) polystyrene (PS), (**b**) 1%Kao-DMSO-PS, and (**c**) 3%Kao-DMSO-PSnanocomposites.

**Figure 11 polymers-15-02094-f011:**
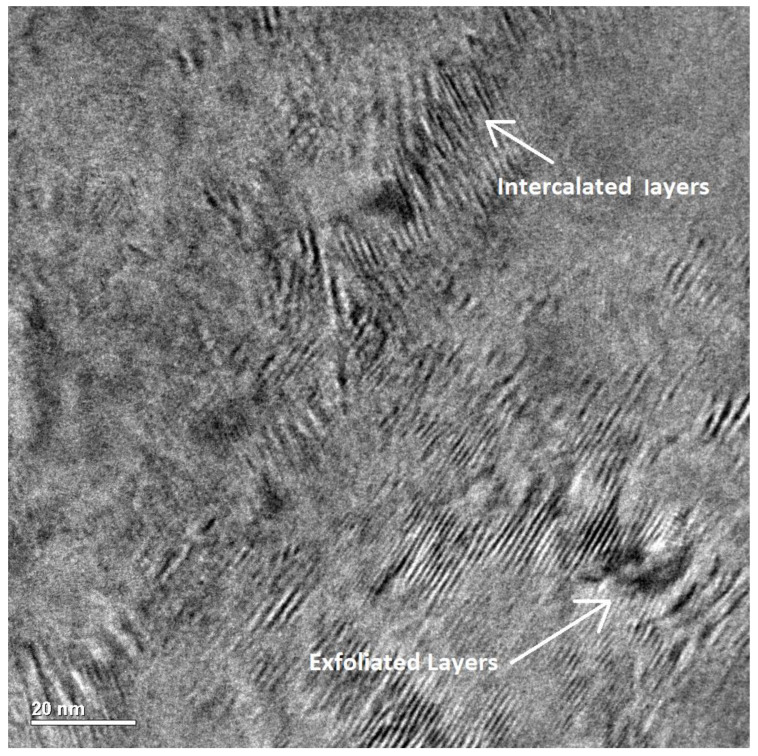
The exfoliated lamellae are shown as having anarc-shaped or anchor morphology. Magnification: 400,000×.

**Table 1 polymers-15-02094-t001:** Residual quantity after polymerization.

Sample	Residue (%)
1%Kao-DMSO-PS	Insignificant
3%Kao-DMSO-PS	Insignificant
5%Kao-DMSO-PS	1.62

## Data Availability

Not applicable.

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
