# Peer review of "The Stability and Properties of Polystyrene/Kaolinite Nanocomposites during Synthesis via Emulsion Polymerization"

_polymers, 2023, doi:10.3390/polym15092094_

Round 1

Reviewer 1 Report

The authors investigated the stability and morphology of the polystyrene/kaolinite latex when incorporated kaolinite nanofiller in detail. By utilizing XRD, FTIR spectrum, SEM and TEM, they analyzed the chemical properties of Kao-DMSO, the distribution of particle size, the nanocomposite conversion and the morphology of exfoliated lamellae. Based on various characterization, they attempted to explain the optimization mechanisms for the polystyrene/kaolinite nanocomposites. While the results are interesting, the manuscript needs to be further revised to address the following comments.

1. There are so many grammar errors in the manuscript, for example:

a) For “The higher particle size (90.39 nm)” in line 145, the word of “higher” should be changed to “larger”.

b) For “can indicate that the monomers migrated from ….” in the line 149, the word of “can” is NOT needed.

c) The description of “The results showed that the addition of 1% of organically modified clay influenced the size of polystyrene particles, increasing it.” is really not understandable. What is the mean of “increasing it”? Why not say “the addition of 1%-weight organic modified clay increased the particle size of polystyrene”?

d) The sentence of “To the reaction time up to 50 minutes is observed,” in line 179 has obvious grammar errors, although I can understand what you really mean.

Here, I wouldn’t like to list the errors one by one, even though there are still many grammar errors in the main text. As known, too many grammar errors will severely limit the communication between authors and readers. So, it is recommended to revise the language for the whole text by a native English speaker.

2. In Figure 3b, no new bands can be found from the description of “the formation of new bands at 3666, 3543, and 3505 cm-1…” in main text. In other words, the mentioned wavenumbers don’t match the wavenumbers marked in the related figure.

3. Please clarify the reason for the decrease in particle size when added 3% clay in the latex, although the authors cited some other research work and tried to make an explanation. In general, high-concentration nanofillers would induce large agglomerates, which will increase the particle size rather than decrease the particle size.

4. In the main text, the author described “From 60 minutes, 1%Kao-DMSO-PS showed a higher conversion than PS and 3%Kao-DMSO-PS.” However, 3% Kao-DMSO-PS provided a higher conversion in comparison with the others for the time range of 50 to 65 in the Figure 7. Please make a precise statement in the main text, maybe you can say “from 70 mins”. Also, please explain the reason for the conversion peak for 3% Kao-DMSO-PS at 60 mins in detail.

5. The authors always attempt to use the explanations derived from other research work to explain their new research results and convince the interested readers. Actually, what the readers expect to learn from this work is the authors’ unique insights about their new results rather than the similar statements from the previous papers. Please provide your own scientific insights for these new results.

6. It is better to mention the percentage of residue for 5% Kao-DMSO-PS in Table 1. What is the method used for evaluating the residue for the others? Since it is hard to say there is no residual in a chemical reaction, especially for the nano-scale polymerization.

7. Please add the scale bar in the Figure 9.

8. In addition to the percentage content of the nanofiller in the latex, the dispersion of nanofiller can also affect the polymerization. In other words, worse dispersion of nanofiller will hinder somehow the polymerization, and then affect the particle size distribution, conversion and morphology for the nanocomposites. How did you avoid the effect of dispersion on the result analysis? Is there any method to evaluate the degree of dispersion of clay in the latex?

Author Response

Reviewer 1: "Please see the attachment."

Reviewer 2 Report

  The aim of this work was to study the stability and morphological properties of polystyrene latex containing kaolinite as filler. The product is prepared during the process of synthesis of nanocomposites by emulsion polymerization. The paper could be considered for publication after the revision. *Why kaolinite was the best as the filler ? * In page 1: 1.2nm)[Error! Bookmark not defined.] and... * How it was decided that 1 wt% of kaolinite is the lowest amount useful for preparation of the polystyrene latex ? * Some useful properties of the polystyrene latex containing kaolinite as filler should be demonstrated in its planned field of application. * What are advantages and disadvantages of the new developed polystyrene latex as compared with that of similar products described already in literature ?

Author Response

Reviewer 2: Please see the attachment.

Round 2

Reviewer 2 Report

Accept in present form